# Social-Demographic Correlates of the Mental Health Conditions among the Chinese Elderly

**Wenjuan Du** [1,2,†], **Jiayi Zhou** [1,†], **Jianjian Liu** [1], **Xuhao Yang** [3], **Hanxu Wang** [4], **Meikun He** [1], **Zongfu Mao** [1,2,*] and **Xiaojun Liu** [1,2,*]

1  School of Health Sciences, Wuhan University, 115# Donghu Road, Wuhan 430071, China; rm002593@whu.edu.cn (W.D.); 2017302180144@whu.edu.cn (J.Z.); jianjianliu@whu.edu.cn (J.L.); 2017203050026@whu.edu.cn (M.H.)
2  Global Health Institute, Wuhan University, 8# South Donghu Road, Wuhan 430072, China
3  Bloomberg School of Public Health, Johns Hopkins University, 615 N. Wolfe Street, Baltimore, MD 21205, USA; xuhao.yang@jhu.edu
4  Cornell Institute for Public Affairs, Cornell University, Ithaca, NY 14850, USA; hw653@cornell.edu
*  Correspondence: zfmao@whu.edu.cn (Z.M.); xiaojunliu@whu.edu.cn (X.L.); Tel.: +86-27-6877-6936 (Z.M. & X.L.)
†  These authors contributed equally to this study.

**Abstract:** Studies on psychological problems among the elderly were mainly conducted in developed countries, which may not fit China under the context of the dramatic changes of social environment. This study aims to assess the status and social-demographic determinants of the mental health among the Chinese elderly. The Chinese version of the Symptom Checklist-90-R (SCL-90-R) was used to measure participants' mental health. A logistic model was established to identify the main socio-demographic factors associated with the overall detection rate of SCL-90-R. The overall positive detection rate of SCL-90-R was 23.6%, and the four symptoms with the highest positive detection rate were somatization (39.5%), obsessive-compulsive disorder (28.1%), other poor mental health symptoms (mainly sleep and diet problems) (25.7%), and depression (25.1%). The results showed those aged 75–79 (OR = 0.640, 95% CI 0.452 to 0.905) and 80 or above (OR = 0.430, 95% CI 0.302 to 0.613), those received 0 (OR = 0.224, 95% CI 0.162 to 0.310) or 1–5 years of education (OR = 0.591, 95% CI 0.449 to 0.776), those were living with spouse only (OR = 0.817, 95% CI 0.563 to 0.997) and with multiple generations (OR = 0.689, 95% CI 0.472 to 0.950), those holding a non-agricultural household registration (OR = 0.727, 95% CI 0.537 to 0.984), and those with an better higher household income were less likely to be positive in overall mental health symptoms. Mental health was shown to be better among those with more advanced ages (≥75), lower levels of schooling (≤5), normal body mass index, higher household incomes, and those who are married and live with their spouse or multiple generations, and those who came from city and currently live in the county.

**Keywords:** mental health; older adults; socio-demographic determinants

## 1. Introduction

The aging of the population affects social and economic development and also exerts instable effects on society. It is estimated that, by 2050, one sixth of the population worldwide will be over 65 years of age (16%), which is twice of that population in 2019 [1]. China has a fast-growing population aging rate. Compared with other countries, the situation of China's aging population appears to be serious. China has the largest number of senior citizens in the world [2]. It is estimated that, by 2020, the number of elderly people over the age of 60 in China will increase to around 255 million, with

its proportion increasing to 17.8% [3,4]. By 2050, the number of elderly people over the age of 80 is expected to reach 90 million [3,4]. It usually takes more than half a century (or even a whole century) for the proportion of the elderly population in developed countries to rise from 7% to 14%, while in China, it only took 30–40 years [5]. Moreover, China faces a structural challenge of individuals becoming old before getting rich [2]. Therefore, China is facing a more serious challenge of population aging than other countries in the world, which has created challenging obstacles on sustainable economic and social development, especially on the implementation of the Healthy China 2030 strategy.

Aging is an inevitable outcome of social development and progress. However, it also causes health concerns for the elderly. With the increase of age, the deterioration of physical function will lead to multiple diseases and physical problems, such as non-communicable diseases [6,7]. To some extent, these diseases have been well controlled due to the development of modern medicine. However, health is not only the absence of disease or pain; it is also the state of mental well-being. Previous studies have shown that mental illnesses such as anxiety [8], depression [9], schizophrenia [10], and emotional disorders [11,12] are becoming more and more serious. The Global Burden of Disease research reported that 20% of the population aged 60 years and over have psychological diseases, with anxiety and depression being the most common [13]. The mental problems of the elderly lead to a heavy burden to the country, including (1) reducing their physical exercise and social activities; (2) directly causing sadness, anxiety, and a worse quality of life; and (3) causing various social problems [14]. In China, due to the implementation of the one-child policy, the transition of the Chinese family structure from multi-child families to single-child families can cause loneliness among the elderly. In addition, social changes caused by industrialization and urbanization may pose a threat to the mental health of the elderly due to their poor social adaptability [15]. Members of the Chinese elderly population are vulnerable to psychological problems under the context of the dramatic changes to the social environment in China. However, mental health is often overlooked in China. Therefore, research on the mental health of the elderly is urgent.

Previous studies from other countries (mostly developed countries) had explored the relationship between general demographic characteristics and the mental health of the elderly [16–18]. However, China's economy is widely acclaimed as a miracle that has brought drastic societal shifts, which exert considerable influence on public health–related issues in recent decades, including mental health–related issues. This results in it being unclear if the findings from developed countries can fit China under the Chinese-specific socio-cultural context, especially because China's aging problem is more complicated than that of other countries. Only by further understanding the association between general demographic characteristics and mental health of the elderly in China can we accurately and effectively implement interventions for the elderly. More population-based studies in China that examine and identify the mental health conditions and characteristics of the elderly population with and their potential socio-demographic correlates could lead to more tailored intervention programs.

Hence, this study aims to understand the current mental health status of the elderly in China and to identify its potential socio-demographic correlates by using the data from the China's Health-Related Quality of Life Survey for Older Adults 2018. It thus may increase social attention toward the mental health of the elderly and provide suggestions for caregivers to put forward targeted interventions in order to improve the physical and mental health of older people and promote the implementation of healthy aging. In addition, it helps to further develop a pension model suitable for the elderly in China and improve the social security system.

## 2. Materials and Methods

### 2.1. Research Design and Participants

The data used in the present study were obtained from a population-based cross-sectional survey, namely, the China's Health-Related Quality of Life Survey for Older Adults 2018. The lead agency was the Global Health Institute of Wuhan University. This survey was conducted in either online or paper format between January and March of 2018. Concerning the uneven level of social and economic development in different regions of China and the high mobility of the population, the Spring Festival holiday is considered to be the best time for a balanced distribution of the country's population. Therefore, collecting data on the socio-ecological factors and health status of the Chinese elderly during this period is the most feasible and valid. The questionnaires were completed by participants themselves or their caregivers according to the elderly people's reading and response abilities. All participants were recruited by convenience sampling.

The China's Health-Related Quality of Life Survey for Older Adults 2018 collected participants' socio-demographic characteristics, personal social capital, health-related behaviors and lifestyles, health-related quality of life, mental health and coping style, etc. Potential participants were the Chinese citizens aged 60 years or older and those who gave informed consent and voluntarily participated in the survey. Those aged 60 and/or over with serious medical conditions, those were at the end of life, those who had been diagnosed with dementia and/or severe mental disorders, and those had lost their daily living abilities or had severe daily living disabilities were excluded from the survey.

The demographic information questionnaire used in this large-scale cross-sectional study was designed based on the China Health and Retirement Longitudinal Study (CHARLS)—a nationally representative survey conducted by the National School of Development of Peking University [19]—and the China Longitudinal Aging Social Survey (CLASS)—a large-scale nationally representative survey conducted by National Survey Research Center at Renmin University of China [20]—with the help of experts and scholars in the field from key universities in China. Since the content of the survey involves many aspects, different scales were used for each aspect. In terms of mental health, the Chinese version of the Symptom Checklist-90-R (SCL-90-R) questionnaire [21] was used. According to the research objectives of this article, we extracted samples of those who completed the SCL-90-R questionnaire from the general database.

### 2.2. Measures

#### 2.2.1. Socio-Demographic Determinants

The social-demographic variables of the participants in this study include sex, age (years), body mass index (BMI), household registration, education (years), current living area, marital status, average annual household income (Chinese Yuan, CNY), living arrangement, and self-reported activities of daily living (ADL) status. Participants were divided into six subgroups based on their age, namely, 60–64, 65–69, 70–74, 75–79, 80–84, and ≥85. The participants were asked to report their height and weight. BMI was calculated as the ratio of weight (kg) to the square of height (m) and were classified as underweight (<18.50 kg/m$^2$), normal (18.50 kg/m$^2$ to 23.99 kg/m$^2$), overweight (24.00 kg/m$^2$ to 27.99 kg/m$^2$), and obese (≥28.00 kg/m$^2$) according to Chinese criteria [22]. The setting of alternative options for participants to answer their educational levels, average annual household income, and current living areas were similar to the CHARLS and the CLASS.

### 2.2.2. Assessment of Mental Health Conditions

The Chinese version of the Symptom Checklist-90-R (SCL-90-R) questionnaire [21] was employed to assess the status of participants' mental health conditions in the present study. The SCL-90-R questionnaire is a commonly used screening tool for diagnosing or measuring the progress and outcome of psychological health conditions. The SCL-90-R can evaluate a broad range of psychological problems and symptoms of psychopathology, including somatization (SOM), obsessive-compulsive symptoms (OCS), interpersonal sensitivity (INTS), depression (DEPR), anxiety (ANX), hostility (HOS), phobic anxiety (PHOA), paranoid ideation (PARI), psychoticism (PSY), and other (mainly reflecting sleep and diet problems). In 1976, Derogatis developed the SCL-90-R questionnaire [23], and the Chinese scholar Zhengyu Wang introduced and edited the Chinese version of SCL-90-R questionnaire in 1984 [21]. Since then, it has been widely used in China to examine the mental health of individuals based on physical and psychological symptoms. The Chinese version of SCL-90-R questionnaire was identified as having the good internal consistency reliability, the its Cronbach's alpha coefficients ranged from 0.78 to 0.96 [21,24,25]. The entire questionnaire contains 90 questions, and all questions use a five-point Likert scale from 0 (not at all) to 4 (extremely). Average scores for individual dimensions that are higher 2 should be considered positive and need further professional diagnosis. Individuals with a total score of 160 points or higher and/or with more than 43 questions out of the 90 questions scored higher than 2 are considered to be positive in the overall situation. Participants took approximately 10 to 15 min to complete the Chinese version of the SCL-90-R questionnaire.

### 2.3. Statistical Analyses

Data was analyzed by using the Statistical Package for the Social Sciences (SPSS) version 23.0 for Windows (SPSS Inc., Chicago, IL, USA). A two-sided *p*-value of less than 5% was considered to be statistically significant. The social-demographic variables of the participants were descriptively analyzed, and frequencies and proportions were reported. The participants' detection rates for nine psychological symptoms and problems obtained from the Chinese version of SCL-90-R questionnaire were calculated as frequencies and proportions, and we also ranked the positive rates of the nine psychological symptoms and problems. Then, we applied chi-square tests to explore the distribution disparities of the overall detection rate among the population with different social-demographic characteristics. Finally, we employed the multivariable logistic model in the modeling analysis to identify the main social-demographic factors associated with overall detection rate of psychological health conditions among the Chinese elderly. The multivariable model included all demographic variables, and model selection was automated. Multivariable adjusted crude odds ratios (OR) value and 95% confidence interval (CI) of the OR were also reported.

### 2.4. Ethical Approval

This study was conducted in accordance with the Declaration of Helsinki, and the study protocol was reviewed and approved by the Institutional Review Board of School of Health Science and Faculty of Medical Sciences, Wuhan University (IRB number: 2019YF2050). Surveys were only conducted if subjects were fully informed of the content and aim of this research project and agreed to participate. The survey was also conducted anonymously, and respondents' information was kept confidential and only for the use of scientific research.

## 3. Results

### 3.1. Demographic Characteristics of the Survey Participants

A total of 2971 individuals was included in the present study, while some values of the variables were missing. Overall, there were slightly more female participants (51.0%), and 14.6% was aged between 75 to 79 years, with 16.4% aged ≥80 years. In terms of individual's body mass index (BMI), 9.6% of the subjects were in the underweight range, while 21.3% were overweight and 3.9% were obese.

Around two-thirds of the sample held an agricultural household registration; nevertheless, 36.1% were currently living in village, and 22.3% in the urban-rural fringe. Among the participants, more than 50% received less than five years of education, and 27.2% were uneducated. As regards average annual household income, 27.5% earned less than RMB 15,000 CNY per year, and 26.1% earned from RMB 15,001–30,000 CNY, which accounted for the majority of the elderly. The survey showed that 28.4% self-reported varying degrees of decline in their activities of daily living (ADL) function. Further details on the demographics information of the survey participants are given in Table 1.

**Table 1.** Demographic information of the survey participants.

| Variables | Sub-Groups | Frequency (*n*) | Valid Percent (%) |
|---|---|---|---|
| Sex | Male | 1441 | 49.0 |
| (28 missing values) | Female | 1502 | 51.0 |
| | 60–64 | 714 | 24.1 |
| Age (years) | 65–69 | 672 | 22.7 |
| (13 missing values) | 70–74 | 656 | 22.2 |
| | 75–79 | 431 | 14.6 |
| | ≥80 | 485 | 16.4 |
| | Underweight (<18.5) | 276 | 9.6 |
| Body mass index | Normal (18.5–23.9) | 1877 | 65.2 |
| (92 missing values) | Overweight (24–27.9) | 613 | 21.3 |
| | Obese (≥28) | 113 | 3.9 |
| Household registration | Agricultural | 1779 | 61.0 |
| (55 missing values) | Non-agricultural | 1137 | 39.0 |
| | 0 | 764 | 27.2 |
| Education (years) | 1–5 | 870 | 30.9 |
| (159 missing values) | 6–8 | 453 | 16.1 |
| | 9–11 | 364 | 12.9 |
| | ≥12 | 361 | 12.8 |
| | Village | 1058 | 36.1 |
| Current living area | The urban-rural fringe | 653 | 22.3 |
| (42 missing values) | County | 627 | 21.4 |
| | Main city zone | 591 | 20.2 |
| Marital status | Married/live together | 1941 | 65.7 |
| (17 missing values) | others | 1013 | 34.3 |
| | ≤15,000 | 805 | 27.5 |
| Average annual | 15,000–30,000 | 762 | 26.1 |
| household income (CNY) | 30,000–45,000 | 665 | 22.8 |
| (48 missing values) | 45,000–60,000 | 406 | 13.9 |
| | ≥60,000 | 285 | 9.8 |
| | Living alone | 315 | 10.7 |
| Living arrangement | Living with spouse only | 1064 | 36.2 |
| (28 missing values) | Living with children | 1028 | 34.9 |
| | Mixed habitation | 536 | 18.2 |
| Activities of daily living | Normal | 2103 | 71.6 |
| (32 missing values) | Decline | 836 | 28.4 |

*3.2. Detection Results of Each Symptom Dimension in SCL-90*

Table 2 displays the amounts and the detection rate for each item of the SCL-90, together with the rank of the positive detection. The positive rate of somatization (39.5%) was among the top of the items, followed by obsessive-compulsive disorder, other (mainly sleep and diet problems), and depression, accounting for 28.1%, 25.7%, and 25.1% of the total sample, respectively. There were 701 individuals detected positive in overall scores, and the overall positive detection rate of SCL-90-R was 23.6%.

**Table 2.** Detection results of the Symptom Checklist-90-R (SCL-90-R).

| Variables | Type | Frequency | Percentage (%) | Rankings |
|---|---|---|---|---|
| SOMA | Negative | 1796 | 60.5 | 1 |
| | Positive | 1175 | 39.5 | |
| OCD | Negative | 2135 | 71.9 | 2 |
| | Positive | 836 | 28.1 | |
| INT | Negative | 2332 | 78.5 | 6 |
| | Positive | 639 | 21.5 | |
| DEPR | Negative | 2226 | 74.9 | 4 |
| | Positive | 745 | 25.1 | |
| ANX | Negative | 2337 | 78.7 | 7 |
| | Positive | 634 | 21.3 | |
| HOST | Negative | 2369 | 79.7 | 8 |
| | Positive | 602 | 20.3 | |
| PHOB | Negative | 2319 | 78.1 | 5 |
| | Positive | 652 | 21.9 | |
| PARA | Negative | 2383 | 80.2 | 10 |
| | Positive | 588 | 19.8 | |
| PSYC | Negative | 2369 | 79.7 | 9 |
| | Positive | 602 | 20.3 | |
| Others | Negative | 2208 | 74.3 | 3 |
| | Positive | 763 | 25.7 | |
| Overall | Negative | 2270 | 76.4 | - |
| | Positive | 701 | 23.6 | |

Notes: SOMA = Somatization, OCD = Obsessive-Compulsive Disorder, INT = Interpersonal Sensitivity, DEPR = Depression, ANX = Anxiety, HOST = Hostility, PHOB = Phobic Anxiety, PARA = Paranoid Ideation, PSYC = Psychoticism.

### 3.3. Distribution of Total Detection Rate by Population Subgroups

The overall detection rate divided by different subgroups are shown in Table 3. The overall positive detection of the SCL-90 was similar between sexes ($\chi^2$ = 0.358, $p$ = 0.549). However, in terms of BMI, 37.7% of the underweight people and 33.6% of the obese people were positive, which was significantly higher than the detection rate of the normal BMI group ($\chi^2$ = 43.608, $p < 0.001$). Individuals with 6–8 years of education, living in villages, and living alone had the highest rate of positive detection, which accounted for 31.3%, 27.9%, and 31.4%, respectively. A significantly higher rate of positive detection was reported among people with an agricultural household registration (26.2%). Yet, those who were currently living in the county (12.8%) reported the lowest overall positive detection of SCL-90 compared with those who were currently living in villages (27.9%), the urban-rural fringe (25.3%), or main city zones (25.2%). Moreover, those with intact ADL function and those who were married or cohabiting showed a lower overall positive detection rate. Those with an average annual household income less than RMB 15,000 CNY had the highest positive detection rate ($\chi^2$ = 115.402, $p < 0.001$).

### 3.4. Factors Associated with Mental Health Conditions among the Chinese Elderly

The parameter estimates for overall detection of SCL-90 together with odds ratios (ORs) and 95% confidence intervals (CIs) by the logistics regression model are illustrated in Table 4. The results showed that the overall detection rate was not statistically different between sexes (OR = 1.099, 95% CI 0.895 to 1.350). When compared with individuals aged 60–64, those aged 75–79 (OR = 0.640, 95% CI 0.452 to 0.905) and those aged 80 or above (OR = 0.430, 95% CI 0.302 to 0.613) were less likely to suffer from mental health problems. Compared with individuals with normal BMI, those who were classified as underweight (OR = 1.640, 95% CI 1.200 to 2.240) and obese (OR = 1.630, 95% CI 1.024 to 2.595) were more likely to be detected positive in overall mental health symptoms. Comparing to people with 6–8 years of education, the odds ratios of positive detection for those with 0 and 1–5 years of education were 0.224 ($p < 0.001$) and 0.591 ($p < 0.001$), respectively. Individuals holding a non-agricultural

household registration had a lower positive detection rate (OR = 0.727, 95% CI 0.537 to 0.984). In terms of current living area, those who were currently living in village (OR = 1.458, 95% CI 1.018 to 2.089) and main urban area (OR = 1.927, 95% CI 1.364 to 2.722) displayed higher rate of detection in contrast to those in county. The results confirmed that those married/cohabiting elderly people had a better mental health condition. With "living alone" as the reference group, people in "living with spouse only" (OR = 0.817, 95% CI 0.563 to 0.997) and "mixed habitation" (OR = 0.689, 95% CI 0.472 to 0.950) categories showed a lower detection rate. Further, elderly people with declined ADL function had a higher rate of detection than those with normal ADL functioning (OR = 3.945, 95% CI 3.170 to 4.911). The odds ratios of positive detection are smaller as the average annual household income increases.

**Table 3.** Distribution of total detection rate by population subgroups.

| Variables | Sub-Groups | Overall Detection | | $\chi^2$ | $p$ |
|---|---|---|---|---|---|
| | | Negative | Positive | | |
| Sex | Male | 1111 (77.1) | 330 (22.9) | 0.358 | 0.549 |
| | Female | 1144 (76.2) | 358 (23.8) | | |
| Age (years) | 60–64 | 541 (75.8) | 173 (24.2) | | |
| | 65–69 | 506 (75.3) | 166 (24.7) | | |
| | 70–74 | 491 (74.8) | 165 (25.2) | 5.006 | 0.287 |
| | 75–79 | 335 (77.7) | 96 (22.3) | | |
| | ≥80 | 387 (79.8) | 98 (20.2) | | |
| Body mass index | Underweight (<18.5) | 172 (62.3) | 104 (37.7) | | |
| | Normal (18.5–24) | 1483 (79.0) | 394 (21.0) | 43.608 | <0.001 |
| | Overweight (24–27) | 466 (76.0) | 147 (24.0) | | |
| | Obese (≥27) | 75 (66.4) | 38 (33.6) | | |
| Education (years) | 0 | 656 (85.9) | 108 (14.1) | | |
| | 1–5 | 673 (77.4) | 197 (22.6) | | |
| | 6–8 | 311 (68.7) | 142 (31.3) | 61.907 | <0.001 |
| | 9–11 | 260 (71.4) | 104 (28.6) | | |
| | ≥12 | 262 (72.6) | 99 (27.4) | | |
| Household registration | Agricultural | 1313 (73.8) | 466 (26.2) | 19.546 | <0.001 |
| | Non-agricultural | 920 (80.9) | 217 (19.1) | | |
| Living area | Village | 763 (72.1) | 295 (27.9) | | |
| | The urban-rural fringe | 488 (74.7) | 165 (25.3) | 53.601 | <0.001 |
| | County | 547 (87.2) | 80 (12.8) | | |
| | Main city zone | 442 (74.8) | 149 (25.2) | | |
| Marital status | Married/Cohabiting | 1512 (77.9) | 429 (22.1) | 115.402 | <0.001 |
| | others | 748 (73.8) | 265 (26.2) | | |
| Average annual household income (CNY) | ≤15,000 | 508 (63.1) | 297 (36.9) | | |
| | 15,001–30,000 | 596 (78.2) | 166 (21.8) | | |
| | 30,001–45,000 | 548 (82.4) | 117 (17.6) | 115.402 | <0.001 |
| | 45,001–60,000 | 346 (85.2) | 60 (14.8) | | |
| | >60,000 | 233 (81.8) | 52 (18.2) | | |
| Living arrangement | Living alone | 216 (68.6) | 99 (31.4) | | |
| | Living with spouse only | 842 (79.1) | 222 (20.9) | 33.932 | <0.001 |
| | Living with children only | 751 (73.1) | 277 (26.9) | | |
| | Mixed habitation | 444 (82.8) | 92 (17.2) | | |
| Activities of daily living | Normal | 1748 (83.1) | 355 (16.9) | 180.752 | <0.001 |
| | decline | 500 (59.8) | 336 (40.2) | | |

**Table 4.** Multiple linear regression model testing the socio-demographic determinants of mental health conditions among the Chinese elderly.

| Variables | OR | 95% CI for OR | | *p* |
|---|---|---|---|---|
| | | Lower | Upper | |
| **Sex (Reference Group = Male)** | | | | |
| Female | 1.099 | 0.895 | 1.350 | 0.369 |
| **Age (years) (Reference group = "60–64")** | | | | |
| 65–69 | 0.916 | 0.688 | 1.220 | 0.549 |
| 70–74 | 0.802 | 0.601 | 1.071 | 0.136 |
| 75–79 | 0.640 | 0.452 | 0.905 | 0.012 |
| ≥80 | 0.430 | 0.302 | 0.613 | <0.001 |
| **BMI (Reference Group = "Normal")** | | | | |
| Underweight | 1.640 | 1.200 | 2.240 | 0.002 |
| Overweight | 1.125 | 0.878 | 1.443 | 0.352 |
| Obese | 1.630 | 1.024 | 2.595 | 0.039 |
| **Education (years) (Reference Group = "6~8")** | | | | |
| 0 | 0.224 | 0.162 | 0.310 | <0.001 |
| 1~5 | 0.591 | 0.449 | 0.776 | <0.001 |
| 9~11 | 1.060 | 0.757 | 1.483 | 0.735 |
| 12 | 1.139 | 0.791 | 1.639 | 0.485 |
| **Household registration (Reference Group = "Agricultural")** | | | | |
| Non-agricultural | 0.727 | 0.537 | 0.984 | 0.039 |
| **Living area (Reference Group = County)** | | | | |
| Village | 1.458 | 1.018 | 2.089 | 0.040 |
| Urban-rural fringe area | 1.385 | 0.977 | 1.965 | 0.068 |
| Main urban area | 1.927 | 1.364 | 2.722 | <0.001 |
| **Marital status (Reference Group= "Married/Cohabiting")** | | | | |
| Others | 1.165 | 1.091 | 1.508 | 0.025 |
| **Average annual household income (CNY) (Reference Group= "≤15,000")** | | | | |
| 15,001–30,000 | 0.509 | 0.394 | 0.657 | <0.001 |
| 30,001–45,000 | 0.426 | 0.314 | 0.579 | <0.001 |
| 45,001–60,000 | 0.317 | 0.217 | 0.465 | <0.001 |
| >60,000 | 0.302 | 0.198 | 0.461 | <0.001 |
| **Living arrangement (Reference Group = "Living alone")** | | | | |
| Living with spouse only | 0.817 | 0.563 | 0.997 | 0.028 |
| Living with children | 1.191 | 0.852 | 1.666 | 0.307 |
| Mixed habitation | 0.689 | 0.472 | 0.950 | 0.035 |
| **ADL (Reference Group = "Normal")** | | | | |
| Decline | 3.945 | 3.170 | 4.911 | <0.001 |

## 4. Discussion

In the present study, there were slightly more females than males, and people holding agricultural registration outnumbered city residents, which accords with China's actual conditions [26]. In addition, the elderly in China tend to be less educated due to historical reasons. Therefore, the sample appears to be representative of the general population.

In this study, the total positive detection rate of SCL-90-R was 23.6%, which is similar to the study conducted in Hebei, China (22.9%) [27]. In this study, among the 10 dimensions assessed by SCL-90-R, the sample obtained a higher rate of detection in the scales of somatization, obsessive-compulsive

disorder, other (mainly reflects sleep and diet problems), and depression. However, a nationwide study carried out in China among individuals aged 18 to 59 years old, found the worse scores in obsessive-compulsive disorder, interpersonal sensitivity, depression, and anxiety [28]. The inconsistency may be attributed to the differences in the study samples. Due to the declines in physiological functioning, elderly people are more vulnerable to somatic dysfunction. Findings from multiple data resources showed that the disability rate of the elderly in China was between 10.48% and 13.31% [29], which, to a large extent, leads to their poor social adaptability and low self-esteem, and thus to mental health problems. Therefore, our study suggests that caregivers should mainly focus on the problems of somatization, obsessive-compulsive disorder, sleep and diet, and depression in the elderly, and establish personalized intervention so as to effectively improve their mental health effectively.

Among the participants, older age was found to be positively associated with a better state of mental well-being, especially for individuals over 75 years old. A population-based cross-sectional study reported similar results [30]. This may be due to the fact that the oldest individuals undertake less family and social responsibility with less stress, while younger elderly need to face the transition of their social roles caused by statutory retirement [31], thus becoming more vulnerable to mental health problems. Marital status and living arrangement are other important factors influencing the mental health of the elderly. Our study showed that single/divorced/widowed persons and those living alone were more likely to suffer from mental problems, which are consistent with the findings from other studies [30,32,33]. In addition, this may be attributed to the changes of family structure from a multi-child family to a single-child one. The only child in the family may bear heavy social responsibilities and financial burdens and has to leave the elderly behind, becoming a migrant worker, which causes loneliness and desolation among left-behind elderly family members. This further confirms the supportive effect of family on the mental health of the elderly.

To a certain degree, our findings are in line with some studies that revealed underweight and obese people have worse mental well-being [34–36]. Existing evidence has indicated that BMI has a significant effect on people's physical health [35], while better physical health increases mental health significantly [37–39]. In our study, individuals with limitations in ADL showed a higher risk of impaired psychological well-being. The possible mechanism might be that physical disability can lead to frustration and the destruction of self-esteem, and is associated with restricted social participation and social networks [40]. Therefore, our study suggests that caregivers should establish prevention strategies to control elderly people's BMI in order to prevent health risks and reduce physical disability.

Similar to earlier findings [30,40], our study shows that the elderly who hold a non-agricultural household registration are less vulnerable to mental problems, which may be related to a more robust health system in cities, since those who hold a non-agricultural household registration have been living in the city all their lives. This result is quite reasonable due to the fact that the social and economic development of cities is far better than that of the countryside. However, in our study, for people who were currently living in the main urban area, their risks of mental disorders were higher than those in the county, which seems to conflict with another study [41]. Actually, China's counties are smaller cities (typically with a population of around half a million to a million), residents in the county also tend to have non-agricultural household registration. Therefore, this result is consistent with previous studies and with the results of household registration variables in this study. What is more, those elderly people who have been living in the county may have more stable social relations, because most of them do not change their living place, causing fewer problems of social adaptability and better mental health.

The main reasons why the mental health status of the elderly who were currently living in the main urban areas was worse may be attributed to China's rapid industrialization and urbanization [42]. The possible explanation of the results are (1) urban residents are more likely to be exposed to environmental contamination, such as air and noise pollution, traffic congestion, which might lead to discomfort, especially for elderly migrants who moved to cities with their working children [43], and (2) the elderly may have difficulties in adapting to fast-pace and high-cost modern life in main urban areas,

thus increasing their stress and dissatisfaction. The result may also be explained by China's unique demographic characteristics. As shown in our study, 61% of the elderly held an agricultural household registration, while only 36.1% currently lived in villages, which shows that some older people in rural areas have migrated to cities due to the migration of their children for work. However, the migration of older people may lead to poor social adaptation due to dramatic differences in modern facilities and life, thus increasing their stress and dissatisfaction. Our study also revealed an adverse effect of living in a village, which is consistent with the observation of worse mental health among people in rural areas [43,44]. This may be due to poor health and welfare systems, fewer entertainment activities, poor living conditions, and lower socio-economic status in rural areas compared with urban places. In addition, due to the population migration of the younger adults to major cities, a great number of elderly people have been left behind in rural areas, leading to their sense of isolation and loneliness [45–47], which has a negative impact on their mental health.

The results of related studies showed that the elderly individuals who received a higher level of education had better mental health since they usually have better socio-economic status, more knowledge about mental disorders, and a more diversified spiritual life [30,43]. However, we found that people with under five years of schooling had a better state of mental well-being. Since the rapid development of society, the socio-cultural environment, and cultural values have resulted in a relatively different context, individuals with higher education are more sensitive to this great change. People with higher education have a more demanding spiritual life and thus may suffer from stress caused by social change, which leads to poor mental health. However, less educated people reflect little on social change. Therefore, they may be less affected. In terms of economic status, our study has shown a positive association between income and mental health, which is consistent with other studies [30,41,44]. It could be explained by their increase in life satisfaction, with better living conditions and more access to health care. Therefore, the government should attach great importance to the economic security of the elderly, increase their social security benefits, ensure their financial stability, and thus promote their mental health.

Although China has attached increasing importance to the mental health of the elderly and released a series of mental health–related prevention and control policy recommendations and guidelines of actions, such as the National Mental Health Plan (2015–2020) [48], there is still a lack of provisions to deal with specific mental problems. This study will become a starting point to draw the public's attention to the mental health conditions for the Chinese elderly population, and we recommend that more attention and research should be focused on mental health issues in the elderly population. Meanwhile, some limitations should be taken into consideration. First, the cross-sectional design does not allow for the identification of causality. Second, the study did not detect some variables that are potentially related to mental health, such as chronic diseases, social capital, and social support. Third, this study only explored the influence of demographics in the general population. We did not provide comparative analyses for distinguishing different groups. Therefore, further studies exploring the mental health of a specific population are needed, especially for the comparison of urban and rural elderly. For the rural elderly, future research needs to pay more attention to the mental health of the left-behind elderly population and the empty-nest elderly population. For the elderly population living in urban areas, future research needs to focus on the elderly population migrating to cities with their children. These limitations should be considered in future studies.

## 5. Conclusions

In conclusion, mental health was shown to be better among individuals at more advanced age (≥75), with lower levels of schooling (≤5), higher incomes, and those who are married and live with their spouse or others. Nonetheless, underweight or obesity, living in the villages or main urban areas, and living alone or having difficulties in performing activities of daily living were all associate with a higher risk of mental health problems. These findings suggest the need for caregivers to establish personalized interventions based on the real condition of the elderly, such as improving their physical

functioning, controlling BMI, and increasing social adaptability. In addition, the government should improve the social security system and create an environment that is supportive of mental health.

**Author Contributions:** Conceptualization, X.L. and Z.M.; methodology, W.D. and X.L.; software, W.D.; validation, J.Z., X.L., and Z.F.; formal analysis, W.D., J.Z., and X.L.; investigation, J.Z., Y.H., M.H., and X.L.; resources, X.L. and Z.M.; data curation, W.D., J.Z., J.L., Y.H., M.H., and X.L.; writing—original draft preparation, W.D., J.Z., J.L., X.Y., and X.L.; writing—review and editing, W.D., J.Z., X.Y., H.W., X.L., and Z.M.; supervision, Z.M.; project administration, X.L. and Z.M.; funding acquisition, Z.M.

**Funding:** This research was funded by Wuhan University "Double First-Class (World's First-Class University & World's First-Class Disciplines)" development (Special fund #C). No support was received from industry. All researchers acted independently of the funding bodies.

**Acknowledgments:** The lead agency of China's Health-Related Quality of Life Survey for Older Adults 2018 is the Global Health Institute of Wuhan University. We would like to express our great appreciation to those students from key universities in China for their help in data collection.

**Conflicts of Interest:** The authors declare no conflict of interest. The funder had no role in the design of the study; in the collection, analyses, or interpretation of data; in the writing of the manuscript, or in the decision to publish the results.

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
