# Peer review of "Social-Demographic Correlates of the Mental Health Conditions among the Chinese Elderly"

_sustainability, doi:10.3390/su11247114_

Round 1

Reviewer 1 Report

I congratulate the authors on an ambitious observational study. The research is robust and the design well considered. I commend the authors for their work - both all of the work leading up to this point and for the planning of this investigation - their contribution Social-Demographic Correlates of the Mental Health in elderly literature. I do have some comments about certain methodological issues covered below under major concerns with the manuscript that require attention prior to publication. These will be discussed below relative to the sections of the manuscript.

TITLE
The title of this manuscript are a little long. Perhaps a more concise version for clarity, interes and ease of read.

ABSTRACT
It is hard to get the detail in an abstract when the word count is limited and this is often the hardest part of a paper to write. However, I do feel that it would be beneficial to explain what specifically you are looking at in relation to mental health (this also applies to the main body of the paper). Is it the development of Social-Demographic Correlates of the Mental Health of pain literature. This needs to be made clearer throughout the paper

KEYWORDS:
Please use recognised MeSH terms as this will assist others when they are searching for information on your research topic. The following website will provide these (simply start typing in a keyword and see if it exists or find an alternative if it does not): https://www.ncbi.nlm.nih.gov/mesh

- INTRODUCTION:
The introduction is weak. An introduction should announce your topic, provide context and a rationale for your work, while catching the reader´s interest and attention. The above has not been given in the introduction that I have read.
Thus, I suggest in this section should be improved, with more details about prevalence, impact related with Social-Demographic Correlates of the Mental Health Conditions among the population elderly. Also, please describe hypothesis in this section.

- MATERIAL AND METHODS: Please, expand and clarification information related with the various questionnaires: 1) China Health and Retirement Longitudinal Study (CHARLS), 2) China Longitudinal Aging Social Survey (CLASS), 3) the Chinese version Symptom Checklist-90-R (SCL-90-R) questionnaire related with reliability and validity and the actual measurements.
- Please describe in the text information related with this research for adhere to reporting STROBE guidelines.
- Lastly, please provide the date of the approved the ethics committee and explain aspects ethics and legal requirement about this research.

- RESULTS: The results in basis of the used method are correct.

- DISCUSSION:
In general the discussion of the results of the study is correct, authors describe the results, the limitations and compare with other researchs.

- CONCLUSION: Novel and interesting study.
FIGURES AND TABLES: Correct.

Author Response

November 30, 2019

Editor-in-Chief

Int. J. Environ. Res. Public Health

Dear Editors and Reviewers,

We are respectfully submitting a revised version of our manuscript, entitled “Social-Demographic Correlates of the Mental Health Conditions among the Chinese elderly: A Population-Based Cross-Sectional Study”, for your consideration for publication in the Section “Human Geography and Social Sustainability”, Special Issue “Aging: Healthcare, Inequalities, Challenges and Trends” of Sustainability.

The authors very much appreciate the thoughtful and critical feedback from the reviewers. We are delighted with your decision. The reviewers are clearly familiar with the topic, as is exemplified by their close and accurate review of this manuscript. The manuscript has been revised according to the reviewers’ comments, and all changes have been highlighted for ready identification. In addition, we have addressed each of the comments from you specifically, and our responses have been outlined in a comment/response format below (see responses to the comments).

I hope the revision is satisfactory, and this manuscript is now acceptable for publication in your journal.

I am looking forward to hearing from you soon.

Sincerely,

Zongfu Mao, Ph.D.

Professor and Director

Global Health Institute

School of Health Science

Wuhan University

115# Donghu Road, Wuhan City 430071,P.R.China

Reviewer 2 Report

Thank you for the opportunity to review this article.

Despite the interesting and actual health-related problem for many countries, there are some issues that are not clarified within the manuscript. Some data seemed to contradict themselves throughout the manuscript.

For example, in Conclusion section is mentioned that ''Mental health was shown to be better among persons with more advanced ages (313 ≧75), lower levels of schooling (≦5) and higher incomes'' (line 312-313). Previous explanations in Discussion section is mentioned that '''we found that people with under five years of schooling had a better state of mental well-being. Since with the rapid development of the society, socio-cultural environment and cultural value are relatively different as compared to the old ones, individuals with higher education may not adapt to the generation gap caused by this great change.'' Less educated people usually problems in adjusting to the society's needs. If they are living in the countryside, where the social/technological demands are not so many, probably this would not be a problem. Bu the authors sustained that there is no differencies between rural and urban individuals. So, was mental health self-evaluated? So, for poor-educated people, this is not a problems? Or was measured by a scale - in this case...the results is strange...

Pleas explain how poor educated people from country side could have higher income. The authors sustained that individuals living in countryside don't have so much expenses. The explanation is not the really one: countryside individuals with poor education have lower incomes. Probably they have less daily living expenses....So, the authors should be very carefully in presenting and explaining the facts.

Authors sustained that people living in cities have worst mental health because of pollution : see lines 272-286:

"In our study, for people who were currently living in the main urban area, their risks of mental disorders were higher than those in county, which was conflicted with other study [41]. The fact may be attributed to China’s rapid industrialization and urbanization [42]. Firstly, urban residents are more likely to be exposed to environmental contamination, such as air and noise pollution, traffic congestion, which might lead to discomfort [43]. Secondly, the elderly may have difficulties in adapting to fast-pace and high-cost modern life in main urban areas, thus increasing their stress and dissatisfaction.

So, if the authors explained this by using air, noise, contamination, pollution....why they constructed that conclusion?

The authors are mentioning that: "Our study also revealed an adverse effect of living in the village, which is consistent with the observation of worse mental health among people in rural areas [43-44]. This may be due to poor health and welfare systems, fewer entertainment activities, poor living conditions and the lower socio-economic status in rural areas compared with urban places. In addition, due to the population migration of the younger adults to major cities, a great number of elderly people have been left behind in rural areas, leading to their sense of isolation and loneliness [45-47], which have a negative impact  on their mental health. Meanwhile, similar to early findings [30,40], our study showed that the elderly who held city registration were less vulnerable to mental problems, which may be related to a morerobust health system in cities''.

Authors mentioned that poor educated people have better mental health. So, the paragraph just cited is not sustaining the conclusion of the research.

Even if the results are well presented, they are not sustained by strong explanations in the Discussion section. So, I recommend to the authors to re-write the Discussion section carefully pointing the logical way of facts.

Please provide a more detailed comparative analysis (gender,  etc)

I think that even if the results are well presented, the interpretation of data, the logical link between the results and other articles' results from the scientific literature is not well done.

Author Response

(The authors gave the same response as above.)

Round 2

Reviewer 1 Report

I think the authors addressed the concerns.

Author Response

Comments and Suggestions for Authors:I think the authors addressed the concerns.

Response: We very much appreciate all the thoughtful and critical feedback from the reviewer.

Reviewer 2 Report

Authors made some improvement for the manuscript.

But some changes are not made yet and arguments still missing for Discussion section.

Please check Table 1 and Table 3 - some variables are the same. Please eliminate those repeating.  Same advice for sections 3.3 and 3.4 Replace words like "ïmmigrants" and "acculturative stress" Discussion section is not supporting by results or some results are nor explained logically by the authors'' explanations.  When explaining the results point take into consideration the following: relationship between BMI and mental and physical status physical and mental health related to incomes.  old people in cities migrated for a job. The less educated you are, lower income you have so negative consequences on mental and physical health explanation that people living in countryside have better health - due to social life and those living in cities are worst due to pollution/traffic and so on.....are not sustained by results. Authors do not collect this kind of information during their research.  ""Therefore, this result is consistent with previous studies and with the results of household registration variables in this study. What is more, those elderly people who have been living in the county may have more stable social relations, most of the elderly people in counties do not change their living place, causing fewer problems of social adaptability and mental health"...not clear (lines 291-294) and so on ....

I recommend to authors to take each results (especially comparative and correlation results) and to explain them using arguments from the research - and present it in congruency or not with other scientific literature

Please insert in Conclusion section exactly the risk factors identify by authors.

Author Response

December 04, 2019

Editor-in-Chief

Sustainability

Dear Editors and Reviewers,

We are respectfully submitting a revised version of our manuscript, entitled “Social-Demographic Correlates of the Mental Health Conditions among the Chinese elderly: A Population-Based Cross-Sectional Study”, for your consideration for publication in the Section “Human Geography and Social Sustainability”, Special Issue “Aging: Healthcare, Inequalities, Challenges and Trends” of Sustainability.

We very much appreciate the thoughtful and critical feedback. The reviewer 1 have requested some further clarification and revisions to our manuscript, which we have completed, and all changes have been highlighted for ready identification. In addition, we have addressed each of the comments from you specifically, and our responses have been outlined in a comment/response format below (see responses to the comments). 

We hope the revision is satisfactory, and this manuscript is now acceptable for publication in your journal.

Looking forward to hearing from you soon.

Sincerely,

Zongfu Mao, Ph.D.

Professor and Director

Global Health Institute

School of Health Science

Wuhan University

115# Donghu Road, Wuhan City 430071,P.R.China
